# Intranasal Therapeutic Peptide Vaccine Promotes Efficient Induction and Trafficking of Cytotoxic T Cell Response for the Clearance of HPV Vaginal Tumors

**DOI:** 10.3390/vaccines8020259

**Published:** 2020-05-29

**Authors:** Gloria Sierra, Stephanie Dorta-Estremera, Venkatesh L. Hegde, Sita M. K. Nookala, Ananta V. Yanamandra, K. Jagannadha Sastry

**Affiliations:** 1UTHealth Graduate School of Biomedical Sciences at Houston, The University of Texas MD Anderson Cancer Center, Houston, TX 77030, USA; ggalvan1@mdanderson.org; 2School of Medicine, University of Puerto Rico, San Juan, PR 00921, USA; stephanie.dorta@upr.edu; 3Department of Thoracic Head and Neck Medical Oncology, The University of Texas MD Anderson Cancer Center, Houston, TX 77030, USA; VLHegde@mdanderson.org (V.L.H.); SMNookala@mdanderson.org (S.M.K.N.); AVYanamandra@mdanderson.org (A.V.Y.)

**Keywords:** human papillomavirus, therapeutic vaccine, adjuvants, intranasal immunization, mucosal immunity

## Abstract

Human papillomavirus (HPV)-induced cancers continue to affect millions of women around the world, and the five year survival rate under the current standard of care for these cancers is less than 60% in some demographics. Therefore there is still an unmet need to develop an effective therapy that can be easily administered to treat established HPV cervical cancer lesions. We sought to investigate the potential of an intranasal HPV peptide therapeutic vaccine incorporating the combination of α-Galactosylceramide (α-GalCer) and CpG-ODN adjuvants (TVAC) against established HPV genital tumors in a syngeneic C57BL/6J mouse model. We obtained evidence to show that TVAC, delivered by the mucosal intranasal route, induced high frequencies of antigen-specific CD8 T cells concurrent with significant reduction in the immunosuppressive regulatory T cells and myeloid derived suppressor cells in the tumor microenvironment (TME), correlating with sustained elimination of established HPV genital tumors in over 85% of mice. Inclusion of both the adjuvants in the vaccine was necessary for significant increase of antigen-specific CD8 T cells to the tumor and antitumor efficacy because vaccination incorporating either adjuvant alone was inefficient. These results strongly support the utility of the TVAC administered by needle-free intranasal route as a safe and effective strategy for the treatment of established genital HPV tumors.

## 1. Introduction

The incidences of human papillomavirus (HPV) driven cancers, including cervical, vulvar, penile, anal, and head and neck cancers continue to increase despite the availability of a prophylactic vaccine targeted at preventing initial infection [1,2,3,4]. This is especially true in countries with limited resources and in need for preventative screening measures. HPV related cervical cancer constitutes one of the leading causes of death in women worldwide with the majority of cases occurring in less developed regions [5]. The current standard of care for cervical cancer is chemo-radiation therapy (CRT), however it can have a drastically negative effect on the overall quality of life for patients, with the 5 year survival rates for some demographics being as low as 58% [6]. Hence, there is an urgent need to develop an effective, easily administered therapy for cervical cancer around the world.

HPV cancers are primarily driven by the sustained expression of the viral proteins E6 and E7. These two proteins are necessary and sufficient for malignant transformation and tumor progression, making them ideal foreign antigen targets for therapeutic cancer vaccines [7]. Although many vaccines targeting these proteins have been developed [8,9,10], none to our knowledge, have proven effective to induce a strong mucosal immune response at the site of HPV infection, the female reproductive tract (FRT). The FRT makes up a part of the highly tolerogenic and compartmentalized mucosal immune system, therefore utilizing an orthotopic tumor model for pre-clinical studies is essential. We reported earlier that intranasal immunization is an effective strategy for the induction of robust immunity against co-administered antigens, as well as for the trafficking of CD8 T cells to the genital mucosa [11,12,13,14]. Using an orthotopic tumor model for pre-clinical studies, we reported that administration of the therapeutic vaccine consisting of HPV16-E6/E7 peptides and α-GalCer adjuvant (TVA) by the intranasal route along with systemic agonistic 4–1BB antibody immune checkpoint therapy (ICT), was curative against vaginally implanted HPV tumors in mice [15]. Although ICT strategies have proven to be effective at combatting various cancers, concerns related to modest success rates, systemic toxicities, and associated cost make them less desirable for low and middle income countries (LMIC) that are greatly affected by HPV-induced cervical neoplasia [16,17,18]. Here, we tested an alternative strategy to ICT by supplementing a second, clinically relevant adjuvant, CpG-ODN to the therapeutic HPV peptide vaccine containing the α-GalCer adjuvant to treat established HPV orthotopic vaginal tumors in mice.

Our results demonstrate that the HPV peptide therapeutic vaccine formulated with α-GalCer and CpG-ODN (TVAC) was effective in inducing sustained tumor regression in the majority of treated mice with a significant survival advantage relative to the vaccine containing either α-GalCer or CpG-ODN adjuvants individually (TVA and TVC, respectively). Importantly, the relative efficacy of TVAC was comparable to that with TVA supplemented with a4–1BB ICT. Tumor regression and survival outcome correlated with robust induction of functional antigen-specific CD8 T cells within the vaginal HPV tumors. These functional CD8 T cells also significantly outnumbered both myeloid derived suppressor cells (MDSCs) and regulatory T cells (Tregs), the major suppressor cell populations typically found in the tumor microenvironment (TME). Our results support TVAC to be a promising approach for treating established HPV driven cervical cancer.

## 2. Materials and Methods

### 2.1. Animals

Female C57BL/6J mice (6–10 weeks) were purchased from the Jackson Laboratories (Bar Harbor, ME, USA) and were maintained in a pathogen-free environment at the institutional animal facility. Animal facilities are fully accredited by the Association for Assessment and Accreditation of Laboratory Animals Care International. All animal procedures were conducted in compliance with the University of Texas MD Anderson Cancer Center Institutional Animal Care and Use Committee (IACUC) guidelines (Project ID: 00000858-RN02, approval date 5/11/2018). Animals were anesthetized with isoflurane for blood draws and hormonal administration, and with a ketamine (100 mg/kg) and xylazine (10 mg/kg) mixture administered by intraperitoneal (IP) route for tumor implantation and N9 Treatment. Animals were euthanized according to IACUC guidelines.

### 2.2. Cell Line and Reagents

The TC-1-luciferase (TC-1–Luc) tumor cell line is a lung fibroblast origin from C57BL/6 mice that was transfected to stably express the E6 and E7 oncogenes of HPV-16 as well as the H-Ras oncogene. This cell line additionally expresses firefly luciferase. This cell line was a kind gift from Drs. T.-C. Wu and C. Hung (Johns Hopkins School of Medicine, Baltimore, MD, USA). Cells were maintained in RPMI 1640 medium supplemented with 10% heat inactivated FBS, 50 units/mL of penicillin–streptomycin, and 50 µg/mL gentamycin.

The E7_44–62_ peptide, Q19 D (QAEPDRAHVYNIVTFCCKCD); E7_49–57_ peptide, R9 F (RAHVYNIVTF); E6_43–57_ peptide, Q15 L (QLLRREVYDFAFRDL); and E6_49–58_ peptide, V10 C (VYDFAFRDLC), were purchased from Elim Biopharma (Hayward, CA, USA) and dissolved in a solution of 10% dimethyl sulfoxide and 90% 1 × PBS at a concentration of 10 mg/mL. The α-GalCer adjuvant was purchased from DiagnoCine (Hackensack, NJ, USA) and dissolved in dimethyl sulfoxide (Sigma, St. Louis, MO, USA) at a concentration of 1 mg/mL. The CpG-ODN 1826 adjuvant was obtained from InvivoGen (San Diego, CA, USA) and dissolved at a concentration of 10 mg/mL in sterile endotoxin free water. Endotoxin-free ovalbumin (OVA) protein was purchased from InvivoGen (San Diego, CA, USA) and reconstituted at 20 mg/mL in PBS. Anti-CD8 blocking antibody was purchased from BioXCell (Lebanon, NH, USA). The α4–1BB monoclonal antibody (LOB12.3) with <1 endotoxin unit/mg of LPS and was purchased from BioXcell (Lebanon, NH, USA) and administered three times (350 ug/dose) via IP route on days 5, 8, and 11 after tumor implantation as previously described [11].

APC-labeled H-2 D ^b^-restricted CD8 T cell epitope E749–57 (RAHYNIVTF)-containing tetramer was procured from the MHC tetramer production facility at Baylor College of Medicine (Houston, TX, USA) and was used for the detection and analysis of antigen-specific CD8 T cell responses in different tissues by flow cytometry.

### 2.3. In Vivo Vaginal Tumor Experiments

For vaginal tumor experiments, female C57BL/6J mice (6–10 weeks) were hormonally synchronized and implanted with 2 × 10^4^ TC-1-Luc cells in the vaginal tract as previously described [19]. Tumor growth was monitored semiweekly using Xenogen In Vivo Imaging System (IVIS) and expressed as Average luminescent signal in select Region of Interest (ROI) (*p*/sec/cm^2^/sr).

### 2.4. Vaccination Treatment

Five days post-vaginal implantation of tumor cells, mice were imaged to ensure successful tumor formation, and were grouped according to size in order to obtain similar average starting tumor size per treatment. Different groups of mice were immunized intranasally with HPV E6/E7 peptides (100 μg each) plus either α-GalCer (2 ug), CpG-ODN (10 μg), or the combination of both adjuvants under ketamine/xylazine anesthesia as previously described [13,14].

### 2.5. Lymphocyte Isolation

For characterization of tumor infiltrating lymphocytes (TIL), mice were implanted with 3 × 10^4^ TC-1–Luc cells in 10 μL PBS/Matrigel mixture at a 2:1 ratio. One week after the second immunization, tumors were harvested, diced, and digested in a mixture of 1 mg/mL collagenase D + DNase for 45 min at 37 °C before being passed through a 45 μm strainer. Lymphocytes were enriched through discontinuous percoll gradient centrifugation and stained for flow cytometry analysis as previously described [15].

For characterization of lymphocytes in the tumors from the female reproductive tract (FRT), the vagina, uterus, and uterine horns were collected and cut into small pieces. Diced tissue was then incubated in 5 mM EDTA for one hour, followed by one hour digestion with collagenase D (1 mg/mL) at 37 °C. Tissues were then passed through a 45 μm strainer and purified by discontinuous percoll gradient centrifugation [11].

### 2.6. CD8 Depletion

For in vivo CD8 depletion, mice were administered 100 ug of aCD8 mAb from BioXCell (Lebanon, NH, USA) via the intraperitoneal (IP) route four days after tumor induction, i.e., one day prior to the first immunization, and every three days after that until the completion of the experiment. Proper CD8 depletion was monitored in the blood throughout the course of the experiment as described in the results section.

### 2.7. Adoptive Transfer of OT-I Cells

The ovalbumin (OVA)-specific OT-I TCR transgenic (Tg) CD8 T cells were obtained from lymph nodes of untreated OT-I mice (CD45.1^+^) and 1 × 10^6^ cells were transferred to congenic C57BL/6J mice (CD45.2^+^) intravenously in 200 uL of sterile PBS. Mice were immunized intranasally with 5 uL of reconstituted endotoxin free OVA alone or in the presence of one or both adjuvants one day after adoptive transfer and sacrificed one week after immunization.

### 2.8. FACS Analysis

Samples were fixed using the Foxp3/Transcription Factor Staining Buffer Set (eBioscience, San Diego, CA, USA) and then stained with antibodies to different surface and intracellular markers obtained from Biolegend (San Diego, CA, USA), BD Biosciences (Franklin Lakes, NJ, USA), eBioscience (San Diego, CA, USA), and Life Technologies (Carlsbad, CA, USA). The CD1 d Tetramer reagent was obtained from NIH tetramer core facility at Emory University (Atlanta, GA, USA). Flow cytometry data were collected on a five-laser, 18-color BD Biosciences LSR II cytometer and analyzed using FlowJo™ Software for Windows, version 10 (Becton, Dickinson and Company, Ashland, OR, USA).

### 2.9. Statistical Analysis

All statistics were calculated using Graphpad Prism version 8 for Windows. Statistical significance was determined using either ordinary one-way ANOVA plus multiple comparisons, or the Brown–Forsythe and Welch ANOVA plus multiple comparisons to test for differences between groups. Statistical significance for survival analysis was calculated using the Mantel–Cox log rank test where indicated. *p* values of <0.05 were considered significant. All figures depict average data values with SEM.

## 3. Results

### 3.1. Therapeutic HPV Peptide Vaccine Containing the Combination of α-GalCer and Cpg-ODN Adjuvants Induces Durable Regression of Established HPV Genital Tumors

We reported earlier that synthetic peptides corresponding to the E6 and E7 oncoproteins of HPV-16 when admixed with α-GalCer adjuvant and delivered by the intranasal route reduced growth of the TC-1–Luc vaginal tumors and that co-administration of agonistic α4–1BB antibody was necessary to induce sustained tumor regression and a significant survival advantage [15]. Because α4–1BB immune checkpoint therapy (ICT) in the clinical setting was reported to be associated with significant toxicity [16,17,18], we tested therapeutic vaccination in the absence of ICT by intranasal administration of the HPV peptides along with α-GalCer and CpG-ODN, two clinically relevant adjuvants with established safety profiles and proven potency to activate dendritic cells through divergent, but complementary, mechanisms [20,21,22]. Groups of C57BL/6J female mice (*n* = 10) were hormonally synchronized as described in the methods section and implanted with 2 × 10^4^ TC-1-Luc tumor cells followed by intranasal administration of the HPV peptide therapeutic vaccine containing α-GalCer and CpG-ODN together (TVAC), α-GalCer alone (TVA), CpG-ODN alone (TVC), or unvaccinated as per the scheme shown in Figure 1A. Prior to the initial vaccination, tumor bearing mice were size matched based on luciferase expression so that the average tumor sizes per group were between 7.36 × 10^2^ and 8.32 × 10^3^ (Figure 1B). Tumor growth was monitored using luciferase expression (Figure 1C) and survival was tracked for 90 days (Figure 1D). We observed sustained tumor regression in a high percentage (~85%) of mice receiving the TVAC (Figure 1C) that resulted in significantly extended survival compared to the untreated group (Figure 1D). Mice treated with TVA or TVC also showed significantly improved survival relative to the untreated control group, but far less percentage of mice exhibiting tumor regression, compared to that in TVAC. More importantly, the protective efficacy afforded by the TVAC in terms of survival advantage was comparable to that observed in mice treated with TVA in combination with α4–1BB antibody (Figure 1D). These results clearly support the efficacy of intranasal therapeutic vaccination, employing the combination of two potent and diverse acting adjuvants to treat established genital HPV tumors without the need for expensive and potentially toxic checkpoint immunotherapy.

### 3.2. Increases in Antigen-Specific and Overall CD8 T Cell Responses Correlate with Efficacy of the Therapeutic HPV Peptide Vaccine Containing the Combination of Adjuvants

Tumor-bearing mice were sacrificed on day 18, one week after the second administration of different treatments and when the tumor growth became representative of the expected outcome (Figure 2A). Tumors were collected from each group of mice and analyzed for different immune cell subsets by flow cytometry. We observed significantly higher percentages of CD8 T cells expressing the cytotoxic molecule Granzyme B (GzmB) in mice treated with TVAC, relative to that of other groups of mice, and this was specifically exemplified in antigen-specific CD8 T cells, detected using an E7 peptide tetramer reagent (Figure 2B). The overall high density of E7 Tetramer + CD8 T cells per milligram of tumor in the TVAC group of mice suggests that generation of functional antigen-specific CD8 T cells and their trafficking to the tumor are major distinguishing correlates of protective immunity (Figure 2C). Furthermore, the greatly expanded percentage of functional CD8 T cells (60.8%) outnumber the total immune inhibitory populations of regulatory T cells (Tregs) and myeloid derived suppressor cells (MDSCs) together (39.2%) within the tumors from mice treated with TVAC relative to those in the other groups of mice (Figure 2D). Figure 2 shows compiled data from three separate experiments with minimum of eight replicates per treatment group.

### 3.3. Antitumor Efficacy of TVAC is Dependent on CD8 T Cell Responses

In order to determine the contribution of the tumor infiltrating CD8 T cells for the TVAC efficacy, in vivo CD8 T cell depletion was performed as described in the methods. Efficacy of the depletion was assessed by monitoring CD8 T cell levels in the blood over time (Figure 3A). The anti-tumor efficacy of TVAC is significantly abrogated in mice receiving the anti-CD8 antibody, but not in the isotype control group in terms of tumor growth and survival (Figure 3B,C, respectively).

### 3.4. Intranasal Vaccination Using α-GalCer and CpG-ODN Adjuvants Induces Significant CD8 T Cell Expansion at the Female Reproductive Tract (FRT)

In order to determine whether the presence of significant populations of antigen-specific CD8 T cells in the tumors at the vaginal mucosa is a function of the adjuvant combination in the vaccine, we adopted the ovalbumin (OVA)-specific OT-I TCR transgenic (Tg) CD8 T cells as a model system. For this, 1 × 10^6^ OT-I cells (CD45.1+) were transferred into congenic wild type C57BL/6J mice (CD45.2+) one day prior to intranasal vaccination with OVA protein alone, or along with either α-GalCer or CpG-ODN, or the two adjuvants together. One week after vaccination we observed dramatic expansion of the adoptively transferred OT-I cells within the spleen and in the FRT of mice vaccinated with OVA plus the two adjuvants, relative to all other groups (Figure 4). These results demonstrate that intranasal vaccination using the combination of the adjuvants α-GalCer and CpG-ODN is an effective strategy to promote robust expansion of antigen-specific CD8 T cells at the FRT.

## 4. Discussion

In this investigation we demonstrate that intranasal immunization with a HPV peptide therapeutic vaccine incorporating the combination of α-GalCer and CpG-ODN adjuvants (TVAC) is an effective approach for eliminating established HPV genital tumors in a syngeneic mouse model. This is particularly relevant for women in countries where a lack of routine cervical cancer screenings are common and access to medical care is limited, because despite attractive preclinical and clinical profiles for several immune checkpoint therapies (ICTs), the overall clinical efficacies have been modest and come with concerns related to high cost of treatment and potential systemic toxicities [23,24]. Our data clearly demonstrates that in the absence of ICT, TVAC exerted superior induction of antigen-specific CD8 T cells resulting in significant increase in the ratios of effector CD8 T cells to the immunosuppressive Tregs and MDSCs within the TME. While CpG-ODN adjuvant is known to induce dendritic cell (DC) activation through Toll-Like Receptor (TLR) 9 signaling, referred to as classical licensing, α-GalCer engages NKT cells to promote strong maturation of DCs, a phenomenon referred to as alternative licensing [22,25]. It is suggested that these adjuvants together have the potency to induce strong activation of naïve CD8 T cells through the activities of two independent sets of chemokine signals [21,26,27]. We believe that the combination of α-GalCer and CpG-ODN adjuvants, relative to each adjuvant alone, in the vaccine enabled promotion of strong antigen-presentation resulting in the induction and recruitment of high levels of antigen-specific CD8 T cells to the tumor. Furthermore, within the tumors of mice treated with TVAC we also observed higher frequencies of functional HPV-specific CD8 T cells in terms of granzyme B expression with the CD8 T cells outnumbering the immunosuppressive Tregs and MDSCs together (Figure 2). This data showing significant induction of functional CD8 T cells is in line with reports in the literature [28]. While the effectiveness of different TLR agonists in combination with α-GalCer administration for enhanced DC-mediated activation of CD8 T cells has been described in vitro using co-cultures of DC and T cells and after subcutaneous delivery in vivo [20,22,29,30,31], our results show for the first time that the needle-free intranasal route of vaccination is efficient to promote induction of functional CD8 T cell responses in systemic and mucosal tissues, and more importantly for significant protective antitumor efficacy.

Because mucosal tissues such as the FRT are more tolerogenic due to their constant exposure to the environmental antigens, it is important to include suitable adjuvants in the vaccine to induce antigen-specific immune responses and this is achieved in our current investigation by the combination of the α-GalCer and CpG-ODN adjuvants in the HPV peptide vaccine [32]. These results are further supported by our earlier studies using the OT-I model where we observed intranasal immunization incorporating the α-GalCer adjuvant induced increased numbers of adoptively transferred OT-I cells in the FRT (30% of CD8+) and other tissues such as spleen (1% of CD8+) [11], and in the present study with the addition of CpG-ODN we noticed a further significant enhancement of antigen-specific CD8 T cells to 60% and 3%, at FRT and spleen, respectively (Figure 4). Additionally, we observed comparable levels of liver enzymes AST and ALT in mice with vaginal HPV tumors that were either treated or not with the HPV peptide therapeutic vaccine incorporating the α-GalCer and CpG-ODN adjuvants individually or as combination; this was in contrast to the use of ICTs such as a4–1BB that have a higher toxicity profile (data not shown), suggesting that a vaccine approach using a combination of adjuvants, particularly α-GalCer + CpG-ODN, against HPV genital tumors is not only an effective but also a safer alternative to ICTs. We have previously reported that intranasal delivery of vaccines is an effective strategy to drive significant levels of antigen-specific CD8 T cell responses at the genital mucosa [11]. Thus, in addition to the effectiveness of the combination of α-GalCer and CpG-ODN adjuvants, our results presented here highlight the importance of the intranasal route of vaccination for increasing antigen-specific CD8 T cell immune responses in the vaginal mucosa for significantly enhancing antitumor efficacy and therapeutic outcome in the genital HPV tumor preclinical model. While combinations of other TLR agonists with α-GalCer have been studied before [29], to our knowledge this is the first report of combining TLR9 agonist with α-GalCer to achieve curative efficacy against HPV genital tumors after intranasal vaccination.

The foreign nature of HPV proteins E6 and E7 make them ideal targets for the development of therapeutic vaccines. Several preclinical studies have reported induction of robust immunity against these proteins using both viral and non-viral vector-mediated immunization schemes, however, their effectiveness was not tested against established orthotopic tumors at the vaginal mucosa [8,10,33]. In the current study, using a HPV peptide therapeutic vaccine employing the combination of adjuvants and intranasal delivery, we could achieve high curative efficacy against established TC-1–Luc vaginal tumors with significant expression of the E6 and E7 oncoproteins in immunocompetent mice with a significant enhancement of functional, HPV-specific anti-tumor CD8 T cell response within the tumors as well as in systemic compartments. It will be interesting to test the effectiveness of this vaccination approach in the context of pre-cancerous lesions, where the expression of the oncoproteins may be lower, resembling the early stages of cervical cancer.

## 5. Conclusions

Here we demonstrate that intranasal immunization with a HPV peptide therapeutic vaccine incorporating the combination of α-GalCer and CpG-ODN adjuvants (TVAC) is an effective approach for eliminating established HPV genital tumors in a syngeneic mouse model. This vaccine is able to effectively increase the percentage of effector and antigen-specific CD8 T cells within the tumors as a successful therapy. In the context of mass-scale immunizations that are important for resource-limited areas of the world, our approach of needle-free intranasal vaccination that is both safe and effective is attractive for the treatment of HPV cancers.

## Figures and Tables

**Figure 1 vaccines-08-00259-f001:**
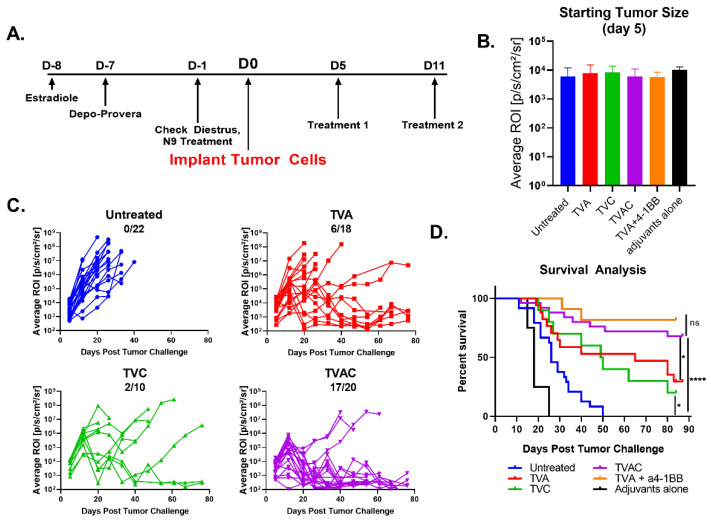
Human papillomavirus (HPV) peptide therapeutic vaccine formulated with the combination of α-GalCer and CpG-ODN adjuvants (TVAC) induces durable regression of established HPV genital tumors. (**A**) Female C57BL/6J mice (*n* = 10 to 22) were hormonally synchronized and challenged with 2 × 10^4^ TC-1-Luc cells into the vaginal cavity. Intranasal immunizations using HPV peptide therapeutic vaccine formulated with either α-GalCer, CpG-ODN, or both α-GalCer and CpG-ODN (TVA, TVC, or TVAC, respectively) were administered on days 5 and 11 after tumor cell implantation; control groups included untreated or immunized mice with the mixture of adjuvants without peptides (adjuvants only). (**B**) Mice were size matched on day 5 prior to immunization based on luciferase expression readout, in terms of ROI units. (**C**) Tumor size was measured using luciferase expression (ROI units). The numbers of mice with complete tumor regression over total per group (minimum 10 mice per group) are shown in each panel for the different groups. (**D**) Survival advantage was recorded between each treatment group as well as the appropriate controls. An additional group of mice receiving intranasal TVA and systemic immunotherapy with agonistic antibody to 4–1BB was included as a positive control based on our earlier published studies for comparing survival rate with that in the TVAC group. Significance in survival proportions was measured using the log-rank test. *p* < 0.05 (*), *p* < 0.00005 (****), ns. = not significant.

**Figure 2 vaccines-08-00259-f002:**
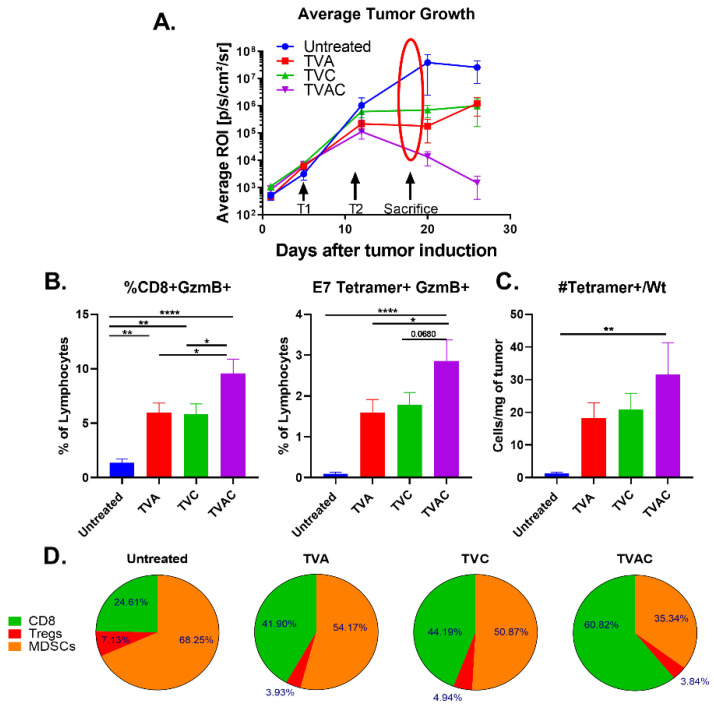
Increases in antigen-specific and overall CD8 T cell responses correlate with efficacy of the therapeutic HPV peptide vaccine. (**A**) Tumor infiltrating lymphocytes (TILs) isolated one week after the last immunization (marked with arrows and the oval) were analyzed by multi-parameter flow cytometry. (**B**) The frequency of granzyme B expressing (GzmB+) total and antigen-specific (E7 tetramer+) CD8 T cells are plotted as percent of live lymphocytes. (**C**) Numbers of antigen-specific CD8 T cells (E7 tetramer+) per milligram of tumor are shown. (**D**) Percentages of tumor infiltrating CD8+T cells, Tregs and myeloid derived suppressor cells (MDSCs) were calculated by dividing each population by the sum of CD8+Treg+MDSC counts per mouse and averaged by treatment group. Statistical significance between treatment groups was calculated via one-way ANOVA. *p* < 0.05 (*), *p* < 0.005 (**), *p* < 0.00005 (****).

**Figure 3 vaccines-08-00259-f003:**
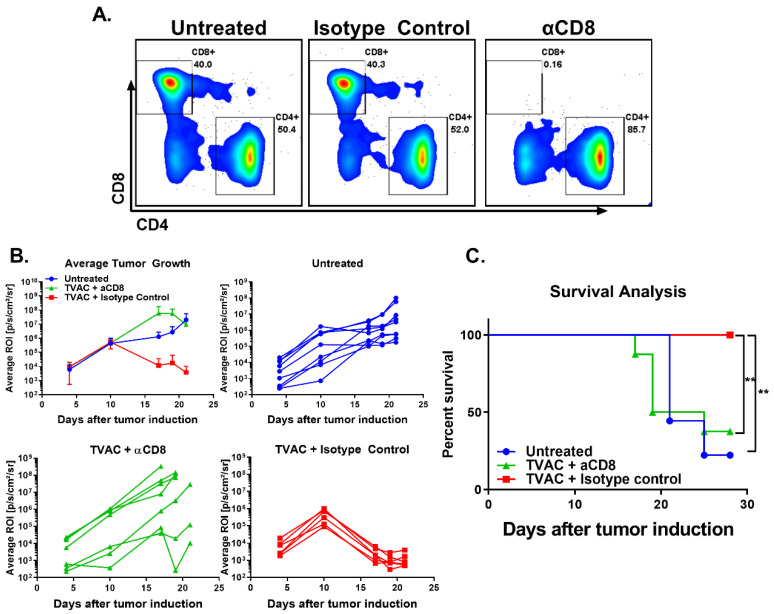
TVAC efficacy is dependent on CD8 T cells. Antibody for the depletion of CD8 T cells was administered by the intraperitoneal route every three days beginning one day prior to tumor induction as described in the methods. Depletion of CD8 T cells in the blood over time was ascertained via flow cytometry and representative flow plots are shown (**A**). Tumor growth and survival were recorded over time (**B**,**C**). Significance in survival proportions was measured using the log-rank test. *p* < 0.005 (**).

**Figure 4 vaccines-08-00259-f004:**
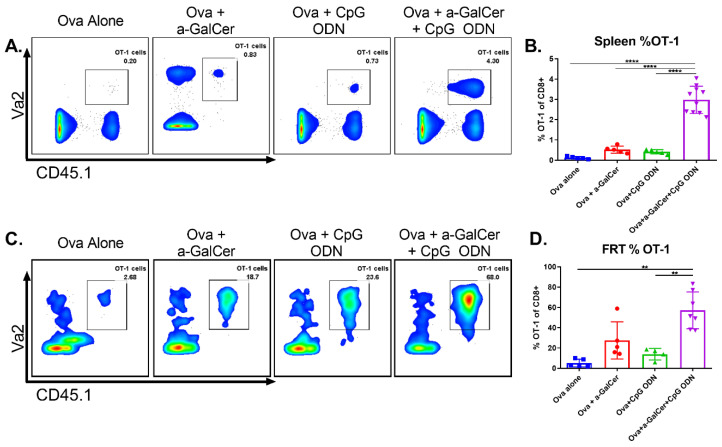
Intranasal vaccination using α-GalCer and CpG-ODN adjuvants affords significant increase of antigen-specific CD8 T cells to the female reproductive tract (FRT). Adoptively transferred OT-I cells were analyzed by flow cytometry in spleen and FRT one week after intranasal ovalbumin immunization using α-GalCer and CpG-ODN adjuvants individually or together. Representative dot plots for spleen (**A**) and FRT (**C**) along with cumulative data from multiple mice in two independent experiments are shown (**B**,**D**). Statistical significance was calculated using ordinary one-way ANOVA with multiple comparisons (B) and the Brown–Forsythe and Welch ANOVA with multiple comparisons (D), *p* < 0.005 (**), *p* < 0.00005 (****).

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
