# Peer review of "Intranasal Therapeutic Peptide Vaccine Promotes Efficient Induction and Trafficking of Cytotoxic T Cell Response for the Clearance of HPV Vaginal Tumors"

_vaccines, 2020, doi:10.3390/vaccines8020259_

Round 1

Reviewer 1 Report

In this manuscript “Intranasal therapeutic peptide vaccine promotes efficient induction and trafficking of cytotoxic T cell response for the clearance of HPV vaginal tumors” by Dr. Sierra and collaborators, a mouse model has been employed to assess the potential of an intranasal HPV peptide therapeutic vaccine (TVAC) incorporating the combination of α-17 GalCer and CpG-ODN adjuvants (TVAC) against established HPV genital tumors.

The main findings of this study indicate that TVAC (administered by intranasal route in female mice) significantly reduced genital tumors in aobut 85% of mice, through a reduction in the immunosuppressive regulatory T cells and myeloid derived suppressor cells in the tumor microenvironment.

Limitations

The number of employed mice/replicates/treatments should be described in the methods section.  

Strengths

The paper is well written, and data are well presented and discussed

The information is presented in a comprehensive way and covers a lot.

With few exceptions, citations cover the relevant literature.

The figures are demonstrative.

In my opinion, this work could be relevant to the field. This is an interesting study with clinical relevance, particularly for the treatment of HPV-related genital tumors. It could be interesting the further evaluation of the efficacy of TVAC in established HPV-driven vulvar/uterine cervical/upper respiratory tract cancers in in vivo models. I have only few specific comments.

Introduction

In this introductive section, other important HPV-driven tumors should be mentioned. Indeed, vulvar cancer (PMID: 32266002), penile/anus cancers (PMID: 21543996) as well as tumors of the upper respiratory tract, including head and neck cancers (PMID: 32362526), have been found to be driven by oncogenic HPV infection. Considering the potential therapeutic role of TVAC against HPV-driven cancers in general, the tumors mentioned above, along with references, should be quoted in the introduction section.

Page 1, line 41 “HPV cancers are primarily driven by the sustained expression of the viral proteins E6 and E7.” Please include references, for instance (PMID: 32322596)

Methods

It is important to clearly indicate in this section the number of animals and replicates employed.

Page 3 lines 128-131. Please indicate in this section after how many days from the tumor induction the CD8T cells depletion was performed.

Results

Page 4 lines 168-174 and page 5 lines 199-202. Please include the p values for every comparison resulted statistically significant different

Page 5 line 198-199: how many mice?

Discussion

Page 8 line 277. “ This data showing significant induction of functional CD8 T cells is in line with reports in the literature.” Please include references

Figures

Fig 2 panel b, graph in the center, the p value >0.05 should be removed.

Reviewer 2 Report

Sierra et al studied a therapeutic vaccine against HPV in mouse model. They found that intranasal administration of HPV E6 and E7 peptides combined with a-GalCer and CpG-ODN adjuvants induced the highest antigen-specific cellular immune responses and suppressed HPV genital tumors. And these correlated with trafficking of antigen-specific CD8 T cells to female reproductive tract.

              The paper is generally well written and structured.

I have not major critiques that can prevent the reviewed work from publication.

Below, some minor suggestions for areas in the manuscript that may deserve improvements.

  1. Fig2, C, D has miss-label (C should be D).
  2. Fig3 A. The letter size of % and CD4, CD8 is too small to be read.
  3. Fig4, A, C. The letter size of % and CD4, CD8 is too small to be read.
  4. There are many studies against E6 and/E7 antigen using viral vector and non-viral vector vaccines. Authors should compare your study to these studies in Discussion section.
  5. Authors should point out the limitation of vaccine, since low expression of E6 and E7 antigen in early stage of cervical cancer.

Reviewer 3 Report

This manuscript described a CD8 T cell-dependent therapeutic effect of a novel intranasal HPV vaccine formulated with alpha-galactosylceramide (aGalCer) and CpG oligodeoxynucleotide (CpG ODN) in C57BL/6 J mice. By administering two doses of vaccine intranasally, a sustained regression of the E6/E7 expressing genital tumor and an increase in survival rate can be observed.

The therapeutic effect of HPV peptide + a-GalCer + CpG ODN is robust and significant, and the accumulation of antigen-specific CD8 T cells in the E6/E7 expressing tumor is convincing. The CD8 T cell depletion experiment demonstrated that CD8 T cells are required for this therapeutic effect.

Overall this manuscript is of excellent quality. The comments and opinions are listed as following:

  • Some of the abbreviations in this manuscript are not clear, such as
    • Line 77 and Line 96. intraperitoneal (IP)
    • Line 68 and Line 267. TME
    • Line 18. a-GalCer and CpG-ODN
  • Line 145. Statistical Analysis.
    • Since the one-way ANOVA test is based on the assumption of Gaussian distribution, the statistical analysis method did not include the normality test. Visually, the "ova+a-GalCer" in Figure 4D may fail the normality test. In addition, the ordinary one-way ANOVA test is based on the assumption of equal variances. Visually, data in Figure 4B may fail the tests for equal variances. The author needs to confirm that all data analyzed by ANOVA test passed both the normality test and Brown-Forsythe Test. If some of the ANOVA tests were the Brown-Forsythe and Welch ANOVA test instead of the ordinary ANOVA test, it would be great to include this information in the legend.
    • It is not clear that the error bars in this paper are Standard Deviation or Standard Error.
  • Line 167. The number of 7.36x10^2 is not consistent with Figure 1B. It should be a typo.
  • Figure 1B.
    • Since TVA+a4-1BB data were included in Figure 1D, the "starting tumor size" of the TVA+a4-1BB group should be included in Figure 1B.
  • Line 205, Line 208, Figure2.
    • There is no Figure 2D.
  • Line 265-267.
    • "TVAC exerted superior two-pronged efficacy in terms of inducing strong antigen-specific CD8 T cells concomitant with a significant reduction in the immunosuppressive Tregs and MDSCs in the TME." Figure 2C did not provide direct evidence that Tregs and MDSCs are reduced. The reduction of Tregs and MDSC in the pie chart (Figure 1C) could be explained by an increase in CD8 T cells with Tregs and MDSC unchanged. Figures of Tregs and MDSCs counts per milligram of the tumor are needed to support the conclusion.
  • Line 272-274.
    • "We believe that the combination of α-GalCer and CpG-ODN adjuvants, relative to each adjuvant alone, in the vaccine enabled promotion of strong antigen-presentation resulting in the induction and recruitment of high levels of antigen specific CD8 T cells to the tumor."
    • This part of the discussion is contradictory to Figure 4. In figure 4, there is no ovalbumin expression in mice FRT; however, OT-I cells still preferentially migrate to the FRT. It suggests that antigen-presentation is not required in the antigen-specific CD8 T cell recruitment. 
  • Result 3.4,Figure4,Line 273, Line287-292.
    • Figure 4 cannot support the conclusion that:
      • "These results demonstrate that intranasal vaccination using the combination of the adjuvants α-GalCer and CpG-ODN is an effective strategy to promote robust expansion of antigen specific CD8 246 T cells as well as their trafficking to the FRT."
      • "These results are further supported by our earlier studies using the OT-I model where we observed intranasal immunization incorporating the α-GalCer adjuvants induced preferential trafficking of the OT-I cells to the FRT (30% of CD8+) over other tissues such as spleen (1% of CD8+) [7], and in the present study with the addition of CpG-ODN we noticed a further significant enhancement of antigen specific CD8 T cells to 60% and 3%, at FRT and spleen, respectively (Fig. 4)."
    • Figure 4 is comparing the percentage of OT-I cells in all CD 8 T cells between the spleen and the female productive tract(FRT). In this comparison, the total CD8 cells are serving as an internal control. This comparison is based on the assumption that the internal control is uniform between the spleen and the FRT. However, this assumption is not correct. Spleen is histologically filled with lymphocytes, while the FRT is not. Because the internal control is not uniform, this comparison would not be proper; therefore, the conclusion is not supported by Figure 4.
    • In fact, Figure 4 can be explained in a general increase in antigen-specific T cells. In a previous study, Intranasal Vaccination Affords Localization and Persistence of Antigen-Specific CD8+ T Lymphocytes in the Female Reproductive Tract, a 17 times increase of antigen-specific OT-1 cells was reported in the blood; thus, all organs without a previous deposit of CD8 cells would expect a considerable increase of antigen-specific T cells over total CD8 T cells, such as a liver or kidney.
    • An additional experiment is required to demonstrate that antigen-specific T cells preferentially traffic to the FRT.

Round 2

Reviewer 1 Report

The authors The manuscript has been accurately revised by the authors according to my comments and it can be accepted for publication as it is. 

Reviewer 3 Report

This manuscript is ready for publication after the latest revision.